# Determination of the Severity and Percentage of COVID-19 Infection through a Hierarchical Deep Learning System

**DOI:** 10.3390/jpm12040535

**Published:** 2022-03-28

**Authors:** Sergio Ortiz, Fernando Rojas, Olga Valenzuela, Luis Javier Herrera, Ignacio Rojas

**Affiliations:** 1School of Technology and Telecommunications Engineering, University of Granada, 18071 Granada, Spain; frojas@ugr.es (F.R.); jherrera@ugr.es (L.J.H.); 2Department of Applied Mathematics, University of Granada, 18071 Granada, Spain; olgavc@ugr.es

**Keywords:** hierarchical intelligent system, deep learning, COVID-19, support vector machine

## Abstract

The coronavirus disease 2019 (COVID-19) has caused millions of deaths and one of the greatest health crises of all time. In this disease, one of the most important aspects is the early detection of the infection to avoid the spread. In addition to this, it is essential to know how the disease progresses in patients, to improve patient care. This contribution presents a novel method based on a hierarchical intelligent system, that analyzes the application of deep learning models to detect and classify patients with COVID-19 using both X-ray and chest computed tomography (CT). The methodology was divided into three phases, the first being the detection of whether or not a patient suffers from COVID-19, the second step being the evaluation of the percentage of infection of this disease and the final phase is to classify the patients according to their severity. Stratification of patients suffering from COVID-19 according to their severity using automatic systems based on machine learning on medical images (especially X-ray and CT of the lungs) provides a powerful tool to help medical experts in decision making. In this article, a new contribution is made to a stratification system with three severity levels (mild, moderate and severe) using a novel histogram database (which defines how the infection is in the different CT slices for a patient suffering from COVID-19). The first two phases use CNN Densenet-161 pre-trained models, and the last uses SVM with LDA supervised learning algorithms as classification models. The initial stage detects the presence of COVID-19 through X-ray multi-class (COVID-19 vs. No-Findings vs. Pneumonia) and the results obtained for accuracy, precision, recall, and F1-score values are 88%, 91%, 87%, and 89%, respectively. The following stage manifested the percentage of COVID-19 infection in the slices of the CT-scans for a patient and the results in the metrics evaluation are 0.95 in Pearson Correlation coefficient, 5.14 in MAE and 8.47 in RMSE. The last stage finally classifies a patient in three degrees of severity as a function of global infection of the lungs and the results achieved are 95% accurate.

## 1. Introduction

The appearance of coronavirus disease 2019 (COVID-19) is a very important turning point within the scope of respiratory diseases. The coronavirus disease 2019 was first detected in the city of Wuhan, Hubei province, China in late 2019 and quickly spread around the world causing devastating effects and millions of deaths. This pneumonia is principally caused by a lung disease driven by a hard acute respiratory syndrome coronavirus type 2 infection [1,2]. It can cause significant damage to patients’ lungs and respiratory system, and has lead to millions of deaths.

Due to this, since the end of 2019, the whole world has been concerned about this important health crisis, which has in the end affected us in many aspects of life. The patients were essentially admitted to hospitals with fever, cough, shortness of breath, and other symptoms. Early detection is essential because infected subjects may not develop symptoms, and be in contact with many other people and transmit it. For these reasons, many research efforts have focused on the detection of COVID-19. The first and the most important method for this is reverse transcription polymerase chain reaction (RT-PCR), which is the most predominantly used method for diagnosing COVID-19 using respiratory samples [3]. The SARS-CoV-2 genes targeted for detection so far include the RdRP gene, Nucleocapsid (N) gene, E gene, Spike protein (S gene), and ORF1ab gene [4]. Reported false negative rates of reverse transcriptase polymerase chain reaction (RT-PCR) is a major problem in disease detection [5]. However, it is not only necessary to know whether or not a patient is infected with COVID-19, it is also very relevant from a medical point of view and for possible therapy, to know the severity of the disease. In this way, medical chest images can be used as an important complement or alternative.

Medical imaging in conjunction with Deep Learning (DL) is a powerful tool to detect and study COVID-19 [6,7]. The automated detection of lung infections in this disease gives the experts great potential to improve diagnosis and complement the information obtained with a primary RT-PCR strategy [8]. Deep learning is accomplished through algorithms which are structurally composed of artificial neurons and multiple data processing layers, in a deep architecture referred to as a Deep Neural Network. In particular, Convolutional Neural Networks (CNNs) have proven to be a very fruitful methodology in image processing and classification [5]. The medical images generally used for this task are X-rays [9,10,11] and computed tomography (CT) images [12,13,14].

Chest radiography is a quick and easy diagnostic test, which is usually requested because of its fast acquisition time and low cost compared to chest CT [15]. However, X-rays have high limitations for detecting COVID-19 infection. For these reasons, and in order to quantify the degree of infection and monitor the evolution of the disease, computed tomography scans are used instead of chest X-rays to evaluate cases of COVID-19 based on the supposed greater sensitivity of the former. Nevertheless, CT scans present higher average acquisition time and cost.

In this contribution, a new intelligent hierarchical methodology with three phases is proposed, for the identification and analysis of the percentage of infection and degree of severity of a patient in the case of COVID-19. In the first step, an automatic classification system is carried out to identify if a patient suffers from COVID-19 disease using chest X-ray images. Once a patient with COVID-19 is detected, a second phase is carried out in order to explore and analyze the percentage of infection in their lungs. This process is done through CT scans of both lungs, which are made up of different slices. The third phase consists of classifying a patient within the same degrees of severity used in clinical practice, according to the percentage of infection of their different lung CT slices.

## 2. Related Word

### 2.1. Classification COVID-19 with X-ray and CT Images

Due to the important role and the powerful feature representation aspects, deep-learning and machine-learning-based approaches are used within computer-aided diagnosis systems in many areas of heath. Since initial outbreak of COVID-19 numerous artificial intelligence investigations have been published on COVID-19. Shambhu et al. realized COVID-19 diagnosis thought a binary Classification of COVID-19 CT Images Using CNN, obtaining an accuracy of 86.9% [16]. Chen Zhao et al. proposed a method for the segmentation and automatic detection of COVID-19 from chest CT images, reaching in segmentation a dice similarity coefficient of 0.9796, a sensitivity of 0.9840 and a specificity of 0.9954, and in the detection an area under the curve (AUC) of 0.9470, a sensitivity of 0.9670 and a specificity of 0.9270 [17]. Feng Shi et al., in their research, distinguished between COVID-19 and pneumonia. The work was divided into two stages, where the segmentation of infection regions was performed in the first stage by CNN VB-Net and then a random forest model was trained to classify, showing a sensitivity of 90.7%, a specificity of 87.2%, and an accuracy of 89.4% [18]. Shuai Wang et al. proposed a deep learning algorithm using CT images to screen for Coronavirus disease, where the CNN proposal was trained using 320 CT images (160 images from COVID-19 negative and 160 images from COVID-19 positive) and 15,000 epochs with an initial learning rate of 0.01. The result of the internal validation reached a total accuracy of 89.5% with a specificity of 88% and sensitivity of 87%. The result of an external testing dataset showed a total accuracy of 79.3% [19]. Hash Pan War et al. 2020 used deep learning for the detection of COVID-19 in X-rays using nCOVnet, and achieved in the test dataset a sensitivity of 97.62% and a specificity of 78.57% [20]. Bougourzi et al. used different architectures ResneXt-50, Densenet-161, and Inception-v3 for COVID-19 Percentage Estimation from CT-Scans, achieving the best performance for slice-level results: 0.9365 in PC, 5.10 in MAE, and 9.25 in RMSE metrics [21]. Loey et al. presented a detection model based on GAN and Deep Transfer Learning, making use of X-ray images. Googlenet, Restnet18 and Alexnet models and four different types of images (COVID-19, normal, bacterial pneumonia and pneumonia viruses) were used in their study to detect the disease. Three stages were considered, where in the first of the four classes used they achieved a test precision of 80.6%, in the second they achieved a test precision of 85.2% and in the last two categories considered (COVID-19 and Normal) they achieved a test accuracy of 100% [22]. Sethy at al. focused on the detection of coronavirus based on deep features and support vector machine using X-ray images achieving 93.4% accuracy [23]. Table 1 shows a summary of different methodologies presented in the bibliography.

### 2.2. Stratifying COVID-19 Patients according to Their Severity

Stratifying patients suffering from COVID-19 according to their severity, using advanced intelligent systems based on medical images, provides a powerful tool for accurate medical information that enables medical experts to support their decision making, optimizes medical resources, and provides great support in characterizing the stages and mechanisms necessary for prevention before patients show severe symptoms.

In the article by Wang et al. [9], a deep learning system for diagnostic and prognostic analysis of COVID-19 using computed tomography is presented. The main goal is to identify possible groups of high- and low-risk patients that can be useful for optimizing medical resources and providing early prevention before patients develop severe symptoms. For the first task, namely diagnosis of patients with COVID-19 from other cases of pneumonia, the performance of the system was AUC = 0.87 and 0.88, respectively, and for viral pneumonia AUC = 0.86. In addition, the deep learning network is good at classifying patients into high and low-risk groups with significant differences in length of hospital stay (*p* = 0.013 and *p* = 0.014, respectively).

Cai et al. [28] developed a random forest model (RF) for classification and regression with the main objective of assessing the severity of disease in COVID-19 patients (categorized into three groups: moderate, severe, and critical) and predicting the patient’s ICU stay, time required for oxygen inhalation, hospital stay, sputum NAT-positive, and patient prognosis. The patient base consists of three groups classified by severity of illness: moderate (*n* = 25), severe (*n* = 47), and critical (*n* = 27), according to clinical staging. To analyze the behavior of RF, the AUC parameter is measured. The prediction for ICU treatment and prognosis (partial recovery vs. prolonged recovery) was 0.945 and 0.960, respectively.

Wang et al. [24] used a DL system with the aim of dividing patients into two groups according to their severity based on CXR images and analyzing the correlation between the DL model and radiologists. The obtained results showed that the Pearson correlation coefficient was 0.81 and the DL system could predict the severity of COVID-19 pneumonia with an AUC of 0.87, indicating the usefulness of the system for experienced radiologists.

In Feng et al. [29], a novel LesionEncoder system is proposed to automatically detect lesions at CT, assess severity, and predict disease progression. The LesionEncoder system consists of a U-Net module and a recurrent neural network (RNN). The authors used CT scans of COVID-19 patient databases in two hospitals in China to train and test the proposed system. Application of the system for patient severity assessment and stratification produced relevant results with a sensitivity of 0.818, specificity of 0.952, precision of 0.940, and AUC of 0.903.

In the work presented by L. Xiao et al. [30], the authors developed a Deep Learning-based model using Multiple Instance Learning and a Residual Convolutional Neural Network (ResNet34) to predict disease severity and further estimate the risk of developing the severe disease in 408 patients suffering from COVID-19 using CT. In the paper by G. Wu et al. [31], a methodology is proposed that combines different sources of information, both clinical variables (information on age, weight, comorbidities, chest tightness, laboratory results, etc.) and information from CT images and physician expert opinions. The goal is to develop an intelligent systems-based model that can predict patients with COVID-19 using the above information to assess the risk of severity during hospitalization and compare performance with the Pneumonia Severity Index (PSI). This index is used by pulmonary specialists to assess the likelihood of severity and mortality in adult patients with community-acquired pneumonia and to assist in the management of hospitalization. Results obtained in [31] showed that patients with COVID-19 who developed severe disease were often older, had multiple comorbidities, had chest tightness, and had abnormal laboratory results and a broader spectrum of lesions in the chest CT on hospital admission. Linear regression was used to analyze the prognosis determined by the system and the expert-reported PSI score. The results for precision are in the range of (0.66–0.95) and for recall in the range of (0.75–0.96). The accuracy ranges from 74.4 to 87.5.

In D. Li et al. [32], a system is created that combines information from medical images (CT, based on neural networks and Deep Learning) together with clinical biochemical indices to develop a prediction model that determines the severity of a COVID-19 patient. A total of 46 COVID-19 patients (10 severe, 36 non-severe) were used in this study. The developed predictive model combining the patients’ radiological results with their clinical biochemical indices achieved an AUROC value of 0.93 and an F1 value of 0.89. The combination of both sources of information (biochemical indices and CT) represents an improvement (approximately 15%) over models based on laboratory test characteristics alone.

In J. Kang et al. [33], a paper is presented using information from the clinical history and biomedical characteristics of the patient and medical images. Cases from the Tumor Center of Union Hospital affiliated with Tongji Medical College in China were used (151 patients). From the original set of 33 clinical variables, six were selected as the most relevant, namely: history of lung disease, age, hemoglobin (Hb), albumin (ALB), globulin (GLB), and blood urea nitrogen (BUN). GLB (r = 0.661, *p* < 0.001) and BUN (r = 0.714, *p* < 0.001) showed strong positive correlation with COVID-19 severity. In addition to this study, a CT imaging analysis of COVID-19 patients was performed. However, this analysis showed a weak correlation (r < 0.2) between imaging manifestations and the severity of patients with COVID-19. This may be because multiple dynamic changes in imaging manifestations need to be considered for the prediction of COVID-19 severity rather than a single imaging manifestation acquired at a specific time point. The model improved good prediction performance (especially GLB and BUN) with an area under the curve value of 0.953 (0.889–0.982).

A mixed model (ACNN) consisting of ANN with clinical parameters and CNN with CT images is also proposed in [34]. The main task is to stratify patients into two groups: low and high risk. A total of 297 COVID-19 patients transferred to five hospitals in Daegu, South Korea, were used. Using the mixed ACNN model, a high classification performance (93.9% accuracy, 80.8% sensitivity, 96.9% specificity, and 0.916 AUC) was obtained with the proposed binary stratification.

## 3. Contribution of the Proposed Novel Methodology

In this paper, a novel method to detect and explore COVID-19 combining both X-ray and CT -scan images is presented. In addition to the detection of COVID by a CNN network using X-ray images, a CNN is used to investigate the degree of infection of each lung slice of the patients with a positive diagnosis using CT scan images. The cohesiveness of the images allows us to examine the impact of the disease on the patient in a different way. Not only is it possible to identify if a person is positive or not, but we can also see the condition of the lung sections. For these pre-trained CNNs in medical images, we are studying how different parameters (optimizers or the learning rate, according to additional work on other related studies) affect the effectiveness of the system.

However, we do not stop at that point, because based on the predictions of the lung slices in the CT images, we use the SVM (supervised classification model) to classify the patient based on the severity of the infection. At this point, we make the greatest contribution by classifying based on the COVID-19 infection severity distribution of patients in a novel way for which there are few precedents in the literature. This is performed by calculating the infection histograms of each patient based on the lung slices. To this end, we are creating a novel database of histograms focused on lung infections, a database that we have pre-labelled based on information about the infected slices. This can be used later and can serve as a starting point for other studies, in addition to classification tasks to locate the infection pattern that may be useful for studying the disease.

In this way and in a very visual way, the infection distribution can be visualized. The proposed novel methodology uses the information from all the CT slices in order to construct the histograms of COVID-19 infection. With these histograms, an SVM with LDA is trained to classify a patient at different states of severity.

## 4. Material: Dataset

In this paper we used two different datasets which are available online. The first dataset is on chest X-ray images (see Figure 1) and the second on CT scans of lungs (see Figure 2).

### 4.1. Dataset 1

The first dataset used was selected from [35] and is composed of X-ray images. This was collected from two different sources and contains images of three classes: COVID-19, Pneumonia, and No-findings.

One of these sources was developed by Cohen JP [36], who provided the COVID-19 X-ray images using various open access sources. There are 127 X-rays with COVID-19 in the database, where 43 belong to females and 82 to males with a positive diagnosis.

The other source is the ChestX-ray8 database provided by Wang et al. [37]. ChestX-ray8 is composed of 108,948 frontal view X-ray images with different pathologies. In this study, and as it is used in [35], to avoid the imbalance data problem and due to the reduced number of COVID-19 images, we used 500 no-findings and 500 pneumonia class front chest X-ray images randomly from this database.

### 4.2. Dataset 2

This dataset is composed of CT scans that were confirmed to have COVID-19. The images were labeled as a function of their infection with COVID-19. This dataset was obtained by [21].

The diagnosis of COVID-19 is based on the most common test, the RT-PCR, and the infection percentage of each slice was identified by two experienced thoracic radiologists. The CT scans were collected from two hospitals: Hakim Saidane Biskra and Ziouch Mohamed Tolga.

Each CT-scan was formed of between 40 and 70 slices. The slices of each CT scan that make up this dataset are those in which the radiologists have the same diagnostic agreement. The number of slices labeled in this set is 3986.

## 5. Methods

The process can be summarized in Figure 3, which shows the architecture of the proposed task. This study can be divided into three stages.

(Stage 1): The first stage detects if a patient is sick with COVID-19 by studying his chest X-ray images. A Densenet-161 network is used, where a training phase uses the X-ray images and different optimizers. The model is developed to provide accurate diagnosis for multi-class classification (COVID vs. Pneumonia vs. No-Findings). This CNN uses cross-entropy as a loss function and the evaluation metrics are classification accuracy, precision, recall and F1-score, where the ground-truth and the predicted labels are compared.

(Stage 2): The second stage detects the infection percentage of COVID-19 by studying CT-scan images. Another Densenet-161 network is used, where the training phase uses the CT-scan images and different optimizers with momentum equal to 0.9. This proposal is evaluated by three evaluation metrics: Mean Absolute Error (MAE), Root Mean Square Error (RMSE) and Pearson Correlation coefficient (PC).

(Stage 3): This stage is carried out after checking the presence of COVID-19 in patients and the percentage of infection associated with this disease in their lungs, in the previous stages. When these percentages are predicted, the samples go through this stage in which the patient is placed into one out of three categories (mild, moderate, or severe grade) using a novel approach based on the histograms of the percentage of infection for each subject. To classify, the Support Vector Machine (SVM) algorithm will be used, studying how the previous use of the Latent Dirichlet Allocation (LDA) method influences the classification.

### 5.1. Densenet

In order to test COVID-19 detection and the infection caused by this disease, we evaluated the performance of two Convolutional Neural Network architectures. The CNN chosen is Densenet-161. DenseNet [38] (Dense Convolutional Network) is a CNN that uses dense connections between the different layers, via Dense Blocks, where they connect the layers (with matching feature map sizes) directly to each other (see Figure 4). An important feature is that the layer receives supplemental inputs from all previous layers and transmits its own feature maps to all subsequent layers. As a consequence, the model is denser and less prone to over-fitting [39].

### 5.2. SVM

This classification method is based on the creation of a data separation hyperplane. This hyperplane performs the function of a divisor, separating the n-dimensional space into two halves where each class of objects is left aside.

A hyperplane can be described by a linear equation of the form:(1)$$w_{1}x_{1}+w_{2}x_{2}+\ldots +w_{n}x_{n}=b.$$

The SVMs will be defined mainly by three factors [40]—Hyperplane, Support Vectors and the Maximum Margin. Therefore, to enable the decision margin to be as wide as possible, the question is to maximize $\frac{2}{\left||w|\right|}$ to obtain the vector *w*, which defines the hyperplane [41].

Because the hyperplane function has *n* variables, and is subject to a series of restrictions, we will use the method of the Lagrange multipliers that solves the problem of constraints, substituting the *z* constraints by *z* variables, thus converting the problem into a case of n+z variables.

### 5.3. LDA

Latent Dirichlet Allocation (LDA) is a mathematical procedure allowing us to minimize the complexity of high dimensional spaces. LDA is a supervised linear dimensionality reduction method stemming from Fisher–Rao LDA [42].

Let X = [$x_{1},\ldots ,X_{N}$] ∈ $R^{DxN}$ be the D-dimensional training data where *N* is the number of total training samples. The class label of the training sample $x_{i}$ ∈ $R^{D}$ is denoted by l($x_{i}$) with l($x_{i}$) ∈ (l, …, C), where C is the number of classes. Denote $N_{i}$ the number of training samples of class *i*. The goal of linear dimensionality reduction is to find a proper transformation matrix W = [$w_{1},\ldots ,W_{d}$] ∈ $R^{Dxd}$ so that *x* ∈ $R^{D}$ is mapped from the D-dimensional space to a low-dimensional space by:(2)$$y=W^{T}x\in R^{d},$$
with *d* < *D*.

As a supervised dimensionality reduction method, LDA aims at finding the optimal transformation vector *w* that maximizes the Rayleigh coefficient:(3)$$J_{LDA}\left(w\right)=\frac{w^{T}S_{b}w}{w^{T}S_{w}w}.$$

### 5.4. Loss Function

The loss function was defined in such a way that making good predictions on the training data is equivalent to having a small loss [43].

Cross-entropy: It is the loss function used in the first Densenet that used X-ray images to predict COVID-19. Cross entropy is used as the training objective, with the purpose of minimizing the distance between the predicted probability scores and the label truth probabilities. The model gives back the probability for each label and, with this, the loss function cross entropy evaluates the model, which is achieved by the following equation:(4)$$L_{CE}=-\sum_{i=1}^{N}p_{i}logq_{i},$$
where $p_{i}$ is the ground-truth percentages, which is equal to 1 if sample *i* is the one belonging to that label, and $q_{i}$ is the predicted percentages for each image. This loss function is minimized using stochastic gradient descent algorithm.

Mean squared error (MSE): It is the loss function used in the second Densenet to predict infection percentage through CT-scans. The MSE function calculates the mean overseen of the squared differences between true and predicted values, which is achieved by the following equation:(5)$$L_{MSE}=\frac{1}{N}\sum_{i=1}^{N}\left(x_{i}-{\hat{x}}_{i}\right),$$
where $x_{i}$ is the ground-truth percentages and ${\hat{x}}_{i}$ is the predicted percentages for each input.

### 5.5. Evaluation Metrics

Accuracy is a ratio of correctly predicted observation to the total observations.
(6)$$Accuracy=\frac{TP+TN}{TP+FP+FN+TN}.$$

Precision is the ratio of correctly predicted positive observations to the total predicted positive observations.
(7)$$Precision=\frac{TP}{TP+FP}.$$

Recall (Sensitivity) is the ratio of correctly predicted positive observations to the all observations in actual class.
(8)$$Recall=\frac{TP}{TP+FN}.$$

F1 score is the weighted average of Precision and Recall. Therefore, this score takes both false positives and false negatives into account.
(9)$$F1Score=\frac{2\times (Recall\times Precision)}{\left(Recall+Precision\right)}.$$

F1 is usually more useful than accuracy, especially if you have an uneven class distribution.

Mean absolute error (MAE) and root mean squared error (RMSE) are two of the most common metrics used to measure accuracy for continuous variables.
(10)$$MAE=\frac{1}{n}\sum_{i=1}^{N}\left|y_{i}-{\hat{y}}_{i}\right|$$
(11)$$RMSE=\sqrt{\frac{1}{n}\sum_{i=1}^{N}{\left(y_{i}-{\hat{y}}_{i}\right)}^{2}},$$
where $y_{i}$ is the ground-truth percentages and ${\hat{y}}_{i}$ is the predicted percentages for each input.

Pearson correlation coefficient (PC) is a statistic that quantifies the relationship between two variables. The variables implied should be continuous in nature.
(12)$$PC=\frac{\sum_{i=1}^{N}\left(y_{i}-{\bar{y}}_{i}\right)\left({\hat{y}}_{i}-{\bar{\hat{y}}}_{i}\right)}{\sqrt{\sum_{i=1}^{N}{\left(Y_{i}-{\bar{Y}}_{i}\right)}^{2}}\sqrt{\sum_{i=1}^{N}{\left({\hat{y}}_{i}-{\bar{\hat{y}}}_{i}\right)}^{2}}}.$$

## 6. Results

The first two stages are based on COVID decisions where each one is developed by a trained Densenet-161 architecture. The Densenet-161 CNNs used are pre-trained in the four identified classes of Lung Disease and Normal from 504 X-ray scans [44]. The third stage uses the prediction of the infectious disease in stage two, to classify the patient as one of three degrees of gravity.

Cross validation (K-fold) is used, where the initial dataset is divided into *k* equitable subsets, of which k−1 sets will be considered to train the classifier and the remaining set will be taken in the evaluation process. The process of choosing the sets is repeated *k* times, taking in each iteration a different set for the classifier test and the remaining k−1 sets for training.

Before using the images on CNN, they are pre-processed in order to enhance the training procedure. All the images are down-sampled to 224 × 224 before being fed to the neural network. Furthermore, two active data augmentation techniques are used—random crop data augmentation and random rotation using an angle between −10 and 10 degrees.

Training batch size is set to 20 and the number of epochs is set to 50 to evaluate the model. The initial learning rate was 1 × 10^3^ for 10 epochs, then the learning rate decreased to 1 × 10^4^ for the next 10 epochs, to 1 × 10^5^ for the next 10 epochs, and finally to 1 × 10^6^ to the end.

### 6.1. Results Stage 1

In this first scenario, we used Densenet-161, where this CNN is re-training by using the X-ray images from dataset 1. Cross-entropy is used as a loss function and this approach is evaluated by classification accuracy, precision, recall and F1-score. The experiments are repeated five times.

For this first stage, the model is fine-tuned for three different optimizers:-SGD optimizer with momentum equal to 0.9.-ADAM optimizer with momentum equal to 0.9.-RMSprop optimizer with momentum equal to 0.9.

To control the multi-class classification performance of the Densenet-161, the model was evaluated in a cross-validation manner, that is, the average classification of five folds is calculated, in order to obtain the global performance.

The confusion matrix (CM) of the different experiments is shown in Figure 5 where we can compare the result of Densenet-161 with different optimizers to detect COVID-19 through X-ray images. The figure shows the summary confusion matrix of the 5-fold validation process, pointing to a better result from the ADAM optimizer, reaching an average accuracy of 88%.

It is worth stressing that if the dataset has an uneven class distribution, it is important to study other parameters in addition to precision to make a good evaluation. For this reason, and focused on CNN with the ADAM optimizer that produces the best result in the first study, precision, recall, and F1-score have also been calculated. The model achieved an average classification accuracy of 88% to classify the different classes and, according to Table 2, the values obtained for precision, recall, and F1-score values are 91%, 87%, and 89%, respectively. We can also look at the results for each k-folds and compare these results. Secondly, the results of confusion matrices in Figure 6 define the distribution of classification imaging in the 5-folds.

### 6.2. Results Stage 2

In this second scenario, we used Densenet-161, where the training phase used the CT-Scans images. Mean squared error was used as a loss function and this proposal was evaluated by three evaluation metrics: Mean Absolute Error (MAE), Root Mean Square Error (RMSE) and Pearson Correlation coefficient (PC).

As the stage 1, in this phase it fine-tuned the model for three different optimizers:-SGD optimizer with momentum equal to 0.9.-ADAM optimizer with momentum equal to 0.9.-RMSprop optimizer with momentum equal to 0.9.

In this stage, following the same procedure as in the previous stage, and to control the performance of the Densenet-161 in the detection of the percentage infection, several metrics were assessed in a cross-validation manner.

Table 3 summarizes the performance of the different aforementioned optimizers. It can be observed that three optimizers obtain roughly the same result, but the RMSprop optimizer produces the best results—0.95, 4.87, 8.10 for PC, MAE and RMSE respectively, the optimizer being selected for the Densenet-161 in this second phase. Figure 7 summarizes the MAE result in the different folds and its evolution with the number of epochs. It can be observed that the initial value of MAE in the first epochs is high because the model is not well adjusted, and when the number of epochs increases and the model learns the patterns, the MAE values decrease until reaching a minimum value of 3.72 in the k-fold 4.

In addition, Figure 8 shows a summary of the best results of the three evaluation metrics used in this stage to evaluate the performance of Densenet-161 with the optimizer selected. We notice in the representation that Fold 4 produced the best results and Fold 3 produced the worst, respectively. This can probably be produced because these folds contain more challenging data than the other folds.

### 6.3. Results Stage 3

As has been mentioned previously, the duty of stage 3 is to classify a patient within three degrees of COVID-19 severity—mild, moderate or severe. For this, a gravity histogram database using all the CT-scans for each patient (Figure 9) was carried out. The process was initiated by the CT-scans dataset2, where the slices belonging to one subject were grouped, and the infection associated to each slice was used to create one histogram for each subject. In this way, the values that are represented in each histogram are the infection percentage of the different slices, the percentage that, in the dataset, was detected by radiologists specialized in COVID-19 infection. It is important to have in mind that not all subjects have the same number of slices on their CT-scan, so all histograms must be normalized with the same factor in order to compare them later. The normalization is performed in such a way that all the added values are given as a whole, that is, they are normalized by the total number of slices of each individual, to remove the dependence on the number of slices.

Thus, each subject is defined by 100 characteristics, each indicating the percentage of slices with the same percentage of COVID-19 infection. For example, if a subject has a value of 0.50 in characteristic 20, this shows that 50% of the slices of CT-scans studied from that patient have 20% infection.

Once the histogram is normalized and the histogram’s dataset is created, a manual labeling of the patients was carried out, since no database has been found that contains on the one hand the infection of each slice, and on the other hand the severity of the total lung volume for all the slices of a subject, simultaneously. As indicated above, each histogram for a patient shows the percentage of slices in each grade of COVID-19 infection ranging from 0% to 100 %. For this reason, the labeling of the histograms was based on the calculation of the mean percentage of infection for each patient. That is, the average infection (AI) is calculated as the sum of the multiplication of the average percentage of slices in each percentage of infection.
(13)$$AI=\sum_{i=0}^{N-1}\left(x_{i}\times y_{i}\right),$$
where $x_{i}$ represents the degree of infection of each slice as a percentage, and $y_{i}$ the normalized number of slices that have that degree of infection. N is the total number of possible degrees of infection, which is equal to 100. It is made clear that $x_{i}$ and $y_{i}$ correspond to the different values of the histogram for each patient. Once the overall average infection (AI) percentage in the lungs of each patient is calculated, labeling is carried out. The first group is the one that has AI<30% all lungs infected, the second is one that is between 30% and 60% infected and the last group are those that are more than 60% infected.

In order to improve the classifier and the prediction of future patients, and taking into account that both the training and the testing of the future classifier will be based on the characteristics associated with the histograms, some of the infection values of the slices of each patient that we have in database2 are going to be slightly modified to create slightly different histograms. Thus, we will have a much larger database with variations of the real database, obtaining more infection histograms with which to train our classifier. Detailing the process a little more, what we do is to assign a different percentage (small modification) of infection to some slices of the patients that we have used in the database and label them as previously commented.

After this, when the patients are labeled as a function of average infection percentage, we use SVM as the algorithm, supervised with 5-folds to train and evaluate this classifier. The classifier input will be the hundreds of characteristics that define a histogram, which, together with their corresponding labels, are what allow the SVM algorithm to know the characteristics associated with each class and to be able to predict the correct label for a new patient.

The results of SVM classification can be observed in Figure 10, which shows the confusion matrix. This representation shows the number of patients that are good or bad, classified as a function of SVM prediction. We can also evaluate the SVM algorithm as a function of parameters such as accuracy, F1-score, precision or recall. These evaluation metrics can be observed in Table 4 and the results are 81% for accuracy, 82% for the average precision, 83% for average recall and 82% for the average F1-score. The term F1-score is important due to the data not being balanced.

To improve the above results, the LDA supervision method was used, allowing us to minimize the complexity of spaces with high dimensionality. LDA allows us to reduce characteristics to N-1, where N is the number of labels. The dimensionality reduction was conducted, taking into account the label to which each subject belongs, since it is a supervised method of dimensionality reduction. For this reason, and having three labels, we changed the representation of each subject from 100 characteristics to two characteristics (denoted X and Y). This can be seen in Figure 11, where each patient is represented by two characteristics. In this way, we managed to reduce the dimensionality, taking into account the degree of severity, thus separating each group of others.

In this way, when we introduce a new patient not evaluated by the pre-trained LDA method, the characteristics will be resized based on the learned patterns. The next step is repeating the SVM classification in order to check if the change in the representation of the characteristics of the patients has an effect on the classification. In Figure 12, the different separation limits created by the SVM classifier can be appreciated.

The confusion matrix of SVM classification, the inputs of which are LDA characteristics, can be observed in Figure 13. This confusion matrix is a summary of the performance of 5 folds.

The evaluation metrics can be observed in Table 5, 95% being the average F1-score, using a total of 596 patients.

Table 6 provides a comparative summary of various alternatives and methods presented in the literature for stratifying COVID-19 patients into different levels of disease severity. The results obtained with the methodology presented in this paper are included.

## 7. Limitations and Practical Applications

One of the most important shortcomings in the application of deep learning in medicine, and focusing on the new disease of COVID-19, is the size of the dataset and the decentralization of this information. When the dataset is too small, the images may have natural features that are too fixed, which could interfere with the model. This effect is responsible for the appearance of a bottleneck and therefore it is recommended that the data be in large amounts (Figure 14). For this reason, and due to the limited number of COVID-19 images, more experiments are needed on a larger set of clearly labeled COVID-19 images for a more detailed estimation of the accuracy of the studies conducted in this paper. It is also important that the data distributions have a similar density for each class, although this can be mitigated by studying not only aspects such as accuracy but also markers such as F1-scores of the isolated classes or by applying up-sampling mechanisms through partial image alteration techniques.

Another of the limitations that is being considered in the implementations is related to the scarcity of additional patient information. The image sets used have the absence of metadata on the pathologies present in the subjects, age, sex or other information that might be necessary to detect possible bias. We consider that not having information about factors such as the pulmonary history of the patients is a great impediment. Some of the patients may have a previous lung problem, in this case COVID-19, and we do not consider this task due to this lack of information. In addition to this, it is also very important to know information that can emphasize the seriousness of COVID-19. Having certain information about the temporal evolution of the disease can also be of great importance for studying how the stage of the disease influences the results of the different models.

In the applications of deep learning models, it is possible to exceed the human precision of many specific tasks within the medical field. However, small variations in the form data acquisition for these models can lead to wrong decisions. This calls into question both the models and their reliability, without being tested for large amounts of data and for highly variable data. Moral and critical questions are also raised about the creation of medical prediction models by people not specialized in the medical subject in question.

For the practical implementation of this research, it is necessary to address and study all the possible external influences that can cause fluctuations in model prediction and explainability, to understand and trust the results and output generated by algorithms. In addition to this and as the main limitation already mentioned, it would be necessary to collect a global database of COVID-19 from different sources and analyze the model and the results in detail. In short, through the work carried out, an attempt is made to provide the community with a new approach from which the study can be continued so that the current COVID detection system can be improved.

## 8. Conclusions

In this paper, a novel approach is presented for the detection of patients with COVID-19 and the classification of these into three different gravity degrees using different stages. For that, two types of lung health imaging are used: X-ray and CT-scans. The study starts with the use of the pre-trained CNN Densenet-161 models that are retrained with the image sets previously mentioned. For this model, the influence of different optimizers is studied in order to improve the performance. In addition, the SVM supervised technique is used to detect characteristics of subjects’ histograms that allow us to classify the patients.

In the first stage, we were able to detect the presence of COVID-19 with a precision of 91%, a recall of 87%, an F1-score of 89% and an accuracy of 88%. These results indicate an encouraging proposal for the detection of COVID-19, which can be used in conjunction with the more widespread technique, PCR. In the second stage, we managed to determine COVID-19 infection in the slices ofCT-scans with results in the evaluation metrics of 0.95 in PC, 5.14 in MAE and 8.47 in RMSE. This indicates that the estimation does not vary in more that 5.14% of infections on average, for each CT-scan slice. Finally, in the third stage we classified one patient in three degrees of gravity as a function of their histograms with results of 95% for F1-score (same percentage for accuracy and recall).

In short, this paper intends to contribute a small grain of sand to the study and detection of COVID-19, which will allow us to control the evolution of the disease. Note that, due to the limited number of COVID-19 images, more experiments are needed on a larger set of clearly labeled COVID-19 images for a more detailed estimation of the accuracy of the studies conducted in this paper. 

## Figures and Tables

**Figure 1 jpm-12-00535-f001:**
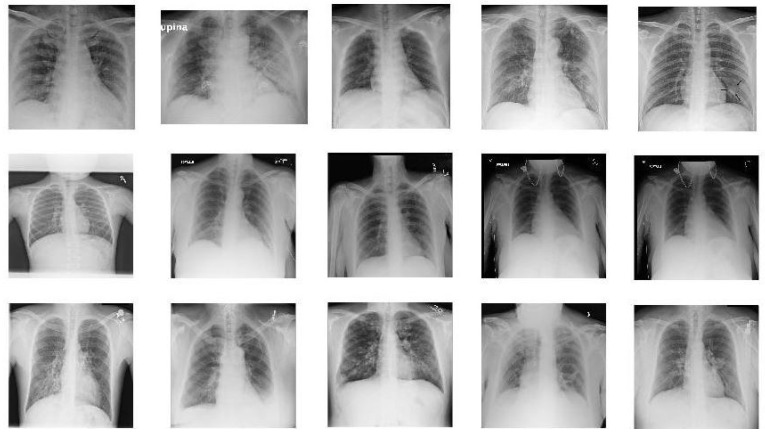
Sample images from X-ray dataset. The images in the first row show 5 COVID-19 images. The images in the second row are 5 sample images of no-finding category in Non-COVID images. The images in the last row give 5 sample images from Pneumonia.

**Figure 2 jpm-12-00535-f002:**
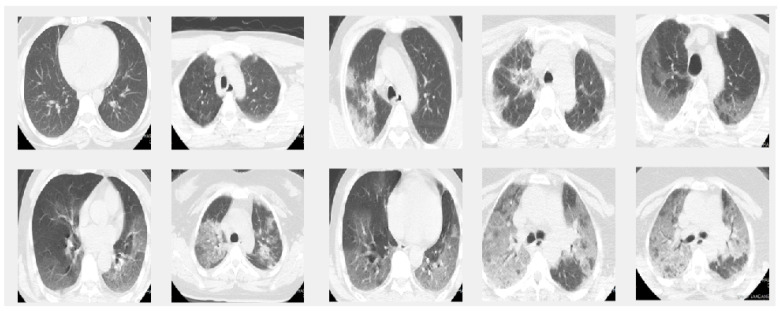
Sample images from the CT dataset. The images in the first row give 5 sample slice images about 0–10–20–30–40% COVID-19 infection in the lungs and the second rows show 50–60–70–80–90% respectively.

**Figure 3 jpm-12-00535-f003:**
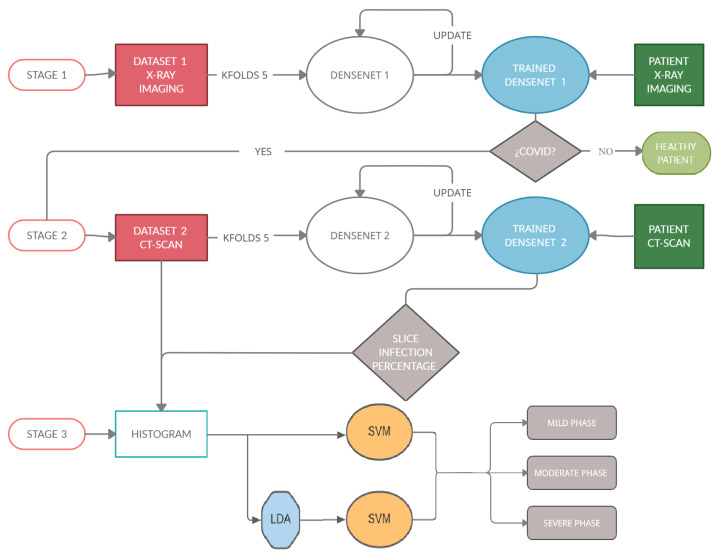
This diagram shows the architecture of the algorithm. (Stage 1): detect if a patient is sick with COVID-19 by studying their chest X-ray images. For this, a Densenet-161 network is used. (Stage 2) detects the percentage of infection in the lungs of a patient with COVID-19. In this stage, another Densenet-161 network is used. (Stage 3) classifies a patient within three degrees of gravity. For this purpose, a novel approach is proposed in which a histogram database is created with patients with different infection percentages. SVM with LDA is used to classify the patients in three degrees of severity.

**Figure 4 jpm-12-00535-f004:**
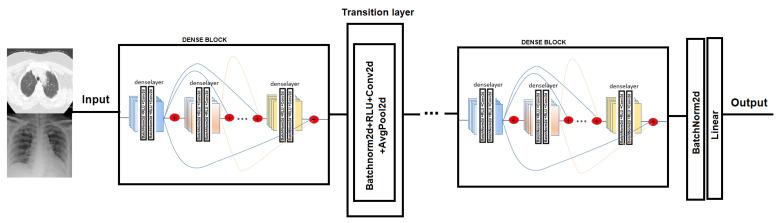
Densenet model network. In the Dense-block each deselayer takes all the preceding feature-maps as an input [38].

**Figure 5 jpm-12-00535-f005:**
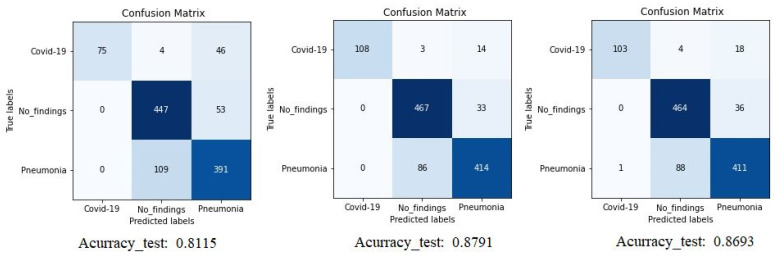
Confusion matrix for Densenet-161 trained with X-ray images. The first matrix is relative to the SGD optimizer, the second is to the ADAM optimizer and the third is to the RMSprop optimizer. (testing phase).

**Figure 6 jpm-12-00535-f006:**
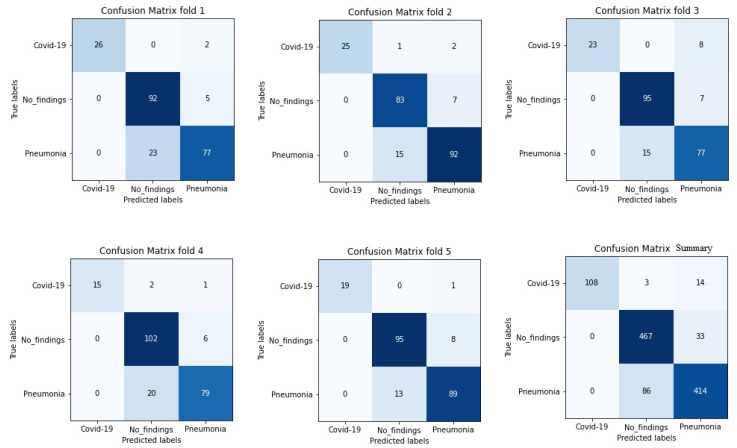
Confusion matrix of 5 folds for Densenet-161 trained with X-ray images and with ADAM optimizer (testing phase).

**Figure 7 jpm-12-00535-f007:**
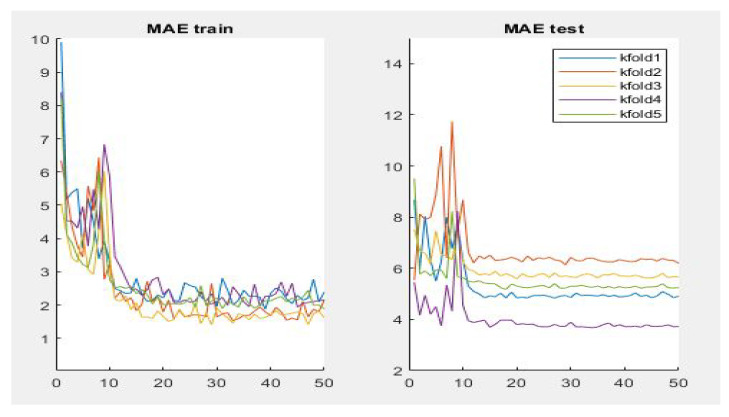
Results of evaluation metric MAE for CNN Densenet-161 with optimizer RMSprop.

**Figure 8 jpm-12-00535-f008:**
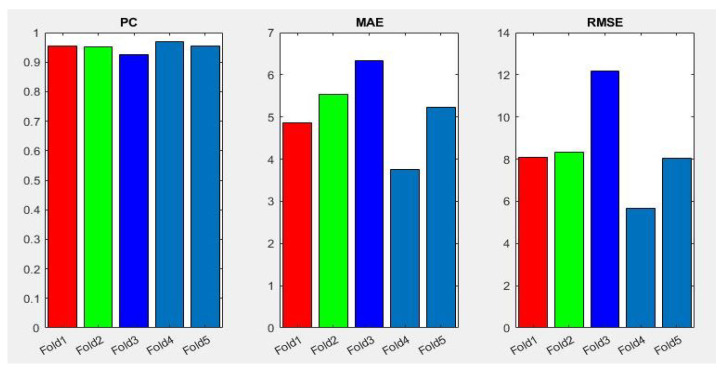
Summary of evaluation metrics results PC, MAE and RMSE for CNN Densenet-161 with optimizer RMSprop (testing phase).

**Figure 9 jpm-12-00535-f009:**
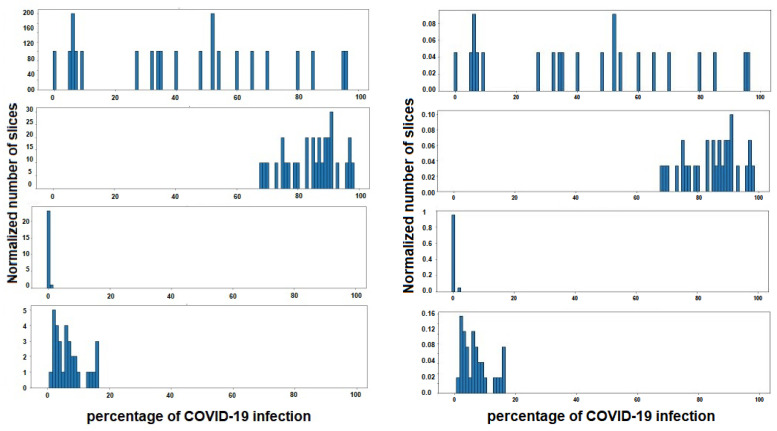
Four instances of histogram dataset. Each histogram belongs to a different subject, and shows the percentage of COVID-19 infections of the different slices of lungs. In the figure on the left the histograms are represented, without normalization, by the number of slices of each patient, and on the right with the normalization applied.

**Figure 10 jpm-12-00535-f010:**
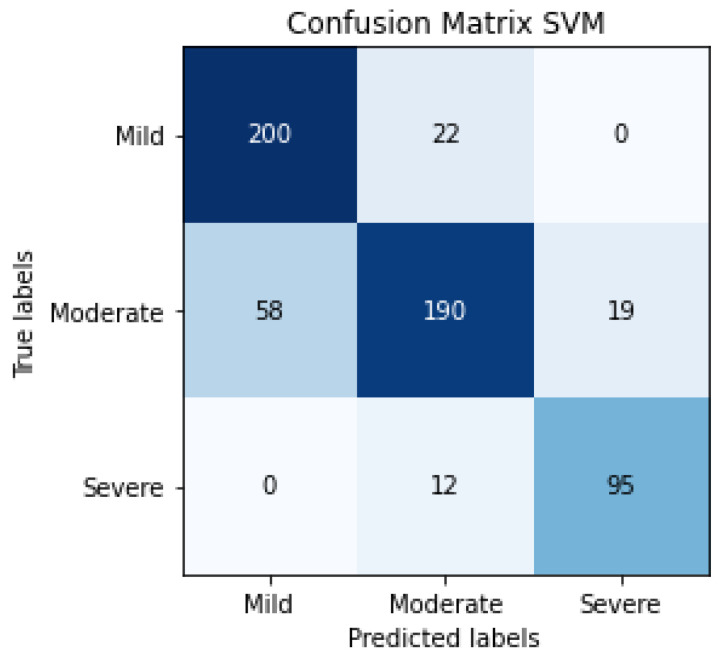
Confusion matrix of the classification in the testing phase using SVM for detecting the severity degree of the lungs of a patient (summary of 5 folds evaluated in testing phase).

**Figure 11 jpm-12-00535-f011:**
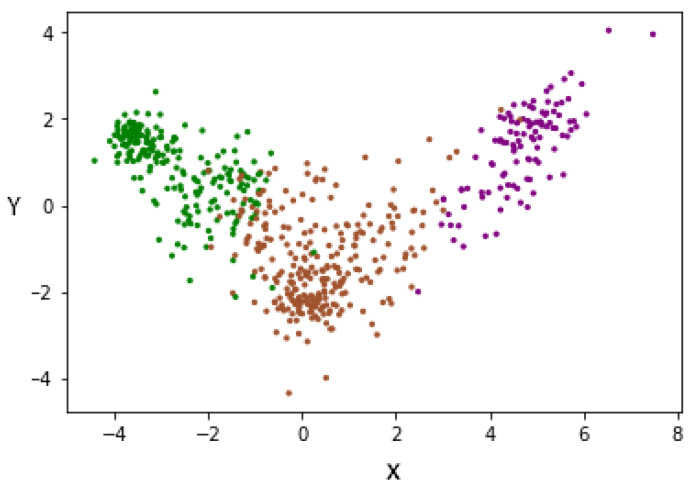
Data redistribution after applying LDA as a supervised dimensionality reduction method. (Green—Mild, Brown—Moderate and Purple—Severe).

**Figure 12 jpm-12-00535-f012:**
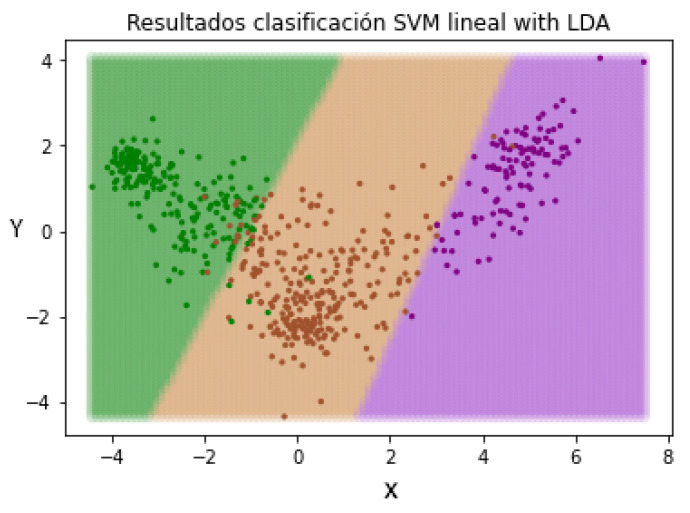
Data redistribution after applying LDA as a supervised dimensionality reduction method. (Green—Mild, Brown—Moderate and Purple—Severe).

**Figure 13 jpm-12-00535-f013:**
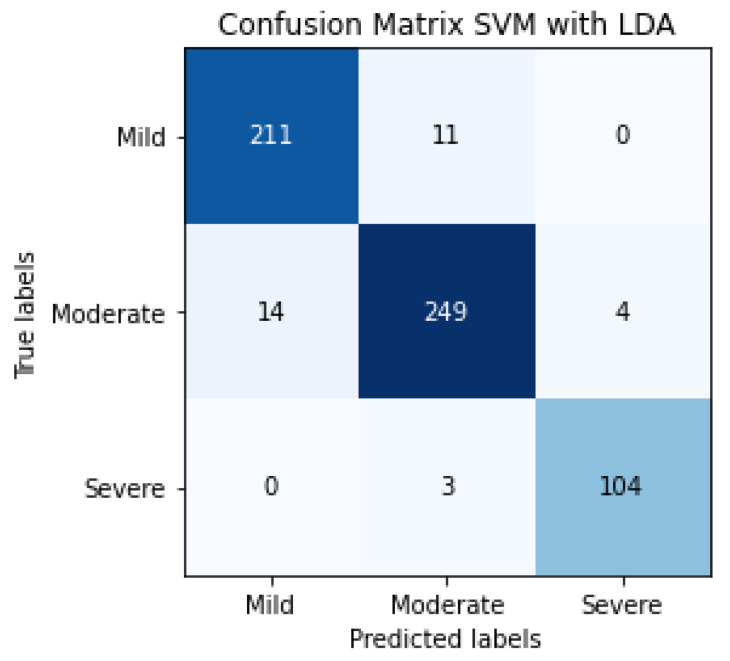
Confusion matrix of the classification in the testing phase using SVM and LDA, for detecting the severity degree of the lungs of a patient (summary of 5 folds evaluated in the testing phase).

**Figure 14 jpm-12-00535-f014:**
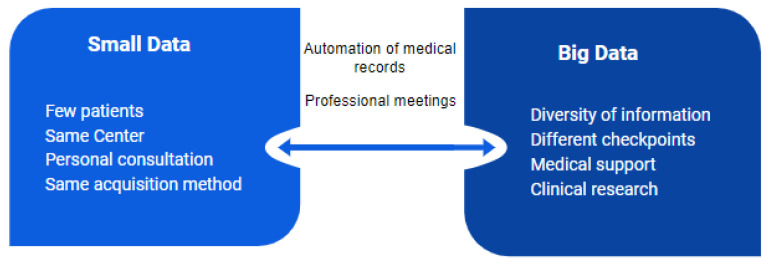
Details of the intersection between small and big data in medicine [48].

**Table 1 jpm-12-00535-t001:** Comparative summary of different methodologies presented in the bibliography.

Author, Year	Objective	Number of Images	Method	Metrics
Shambhu et al., 2021 [16]	COVID-19 diagnosis (CT images)	COVID-19 = 349 Normal = 397	CNN	Accuracy = 86.9%
Chen Zhao et al., 2021 [17]	Segmentation and automatic detection of COVID-19 (CT images)	COVID-19 = 58 CAP = 24 Normal = 30	SP-V-Net	AUC = 94.7%, sensitivity = 96.7%, specificity = 92.7%
Feng Shi et al., 2021 [18]	COVID-19 Detection (CT images)	COVID-19 = 1.658 Pneumonia = 1.027	VB-Net random_forest	sensitivity = 90.7%, specificity = 87.2%, accuracy = 89.4%
Shuai Wang et al., 2021 [24]	Screen for Corona virus disease (CT images)	COVID-19 = 160 Normal = 160.	Covid-net	Validation: sensitivity = 87%, specificity = 88%, accuracy = 89.5%. Testing: accuracy = 79.3%
PanWar et al., 2020 [20]	COVID-19 Detection (X-rays)	COVID-19 = 192 Normal = 145	nCOVnet	sensitivity = 97.62%, specificity = 78.57%
Bougourzi et al., 2021 [21]	COVID-19 Percentage Estimation (CT images)	COVID-19 = 3986	ResneXt-50, Densenet-161, Inception-v3	PC = 0.9365, MAE = 5.10, RMSE = 9.25.
Loey et al., 2020 [22]	COVID-19 Detection (X-ray images)	COVID-19 = 69 Pneumonia_bac = 79 Pneumonia_vir = 79 Normal = 79	Googlenet, Restnet18, Alexnet	Four-classes: precision = 80.6% Three-classes: precision = 85.2% Two-classes: precision = 100%
Sethy et al., 2020 [23]	COVID-19 Detection (X-ray images)	COVID-19 = 25 Normal = 25	ResNet50 SVM	ResNet50-SVM: accuracy = 95.38%, SVM: accuracy = 93.4%
Wang el al., 2020 [9]	COVID-19 Detection (X-ray images)	COVID-19 = 266 Normal = 358	Covid-net	sensitivity = 91% and VPP = 98.9%
Arifin et al., 2021 [25]	Fast COVID-19 Detection (X-ray images)	COVID-19 = 50 Pneumonia = 50 Normal = 50	MobileNet-V1 MobileNet-V2	accuracy = 93.24% Single-Shot MobileNet-V1: accuracy = 83.7%. MobileNet-V2: accuracy = 87.5%
Huang et al., 2020 [26]	Quantitative Chest CT Assessment of COVID-19	COVID-19 moderate = 94 severe = 20 critical = 6	CNN	opacification percentage (8.7% (2.7%, 21.2%) vs. 6.0% (1.9%, 24.3%); P = 0.655)
Shan et al., 2020 [27]	Lung Infection Quantification of COVID-19 (CT Images)	COVID-19 = 300	VB-Net	DSC = 91.6% ± 10.0% POI = 0.3%
Proposed methodology	COVID-19 Detection (X-ray images)	COVID-19 = 50 Pneumonia = 500 Normal = 500	Densenet-161	Accuracy = 88% precision = 91%, recall = 87% F1-scored = 89%
Proposed methodology	COVID-19 Percentage Infection (CT images)	COVID-19 = 3986	Densenet-161	PC = 0.95, MAE = 5.14, RMSE = 8.47.

**Table 2 jpm-12-00535-t002:** Precision, Recall, F1-score and Accuracy of Densenet-161 model with X-ray images and with ADAM optimizer.

Fold-1	Precision	Recall	F1-Score	#Images
COVID-19	100	93	96	28
No-findings	80	95	87	97
Pneumonia	92	77	84	100
Accuracy			87	225
**Fold-2**	**Precision**	**Recall**	**F1-Score**	**#Images**
COVID-19	100	89	94	28
No-findings	80	92	88	90
Pneumonia	91	86	88	107
Accuracy			89	225
**Fold-3**	**Precision**	**Recall**	**F1-Score**	**#Images**
COVID-19	100	74	85	31
No-findings	80	93	90	102
Pneumonia	92	84	84	92
Accuracy			87	225
**Fold-4**	**Precision**	**Recall**	**F1-Score**	**#Images**
COVID-19	100	83	91	18
No-findings	88	94	88	108
Pneumonia	92	80	85	99
Accuracy			87	225
**Fold-5**	**Precision**	**Recall**	**F1-Score**	**#Images**
COVID-19	100	95	97	20
No-findings	88	92	90	103
Pneumonia	91	87	89	102
Accuracy			90	225
**Summary**	**Precision**	**Recall**	**F1-Score**	**#Images**
COVID-19	100	86	93	125
No-findings	84	93	88	500
Pneumonia	90	83	86	500
Accuracy			88	1125

**Table 3 jpm-12-00535-t003:** Results of the second stage in the testing phase using CNN Densenet-161, with three different optimizers and loss functions MSE.

SGD Optimizer			
	PC	MAE	RMSE
K-fold-1	0.95	5.07	8.37
K-fold-2	0.92	7.25	10.03
K-fold-3	0.92	5.64	11.57
K-fold-4	0.97	3.88	6.03
K-fold-5	0.94	5.71	8.75
Summary	0.94	5.51	8.95
**RMSprop Optimizer**			
	**PC**	**MAE**	**RMSE**
K-fold-1	0.96	4.87	8.10
K-fold-2	0.95	5.54	8.34
K-fold-3	0.92	6.33	12.16
K-fold-4	0.97	3.75	5.69
K-fold-5	0.95	5.22	8.05
Summary	0.95	5.14	8.47
**ADAM Optimizer**			
	**PC**	**MAE**	**RMSE**
K-fold-1	0.95	5.27	8.35
K-fold-2	0.90	6.36	11.60
K-fold-3	0.92	5.84	11.66
K-fold-4	0.97	3.87	5.80
K-fold-5	0.95	5.21	8.24
Summary	0.94	5.31	9.13

**Table 4 jpm-12-00535-t004:** Results of the classification in the testing phase using SVM for detecting the severity degree of the lungs of a patient (summary of 5 folds evaluated in the testing phase).

SVM Results (%)				
	Precision	Recall	F1-Score	#Patients
Mild	78	90	83	222
Moderate	85	71	77	267
Severe	83	89	86	107
Accuracy			81	596

**Table 5 jpm-12-00535-t005:** Results of the classification in the testing phase using SVM and LDA for detecting the severity degree of the lungs of a patient (summary of 5 folds evaluated in the testing phase).

SVM with LDA Results (%)				
	Precision	Recall	F1-Score	#Patients
Mild	94	95	94	222
Moderate	95	93	94	267
Severe	96	97	97	107
Accuracy			95	596

**Table 6 jpm-12-00535-t006:** Performance comparative of different methodologies presented in the bibliography to predict disease severity of patients with COVID-19.

Author	Methods	Information Used	N° Patients (COVID+)	Results
Feng [29]	Recurrent Neural Network. LesionEncoder framework	CT	347	Recall = 0.81; AUC = 0.90; Accuracy = 94%
Cai [28]	Random Forest	CT/laboratory	99	AUC = 0.945
Xiao [30]	Convolutional Neural Network (ResNet34)	CT	408	AUC = 0.89; Accuracy = 82%
Wu [31]	Linear regression	CT/laboratory	725	Precision = (0.66–0.95); Recall = (0.75–0.96); AUC = (0.84–0.93); Accuracy = (74%–87%)
Li [32]	Convolutional Neural Network	CT/laboratory	46	Precision = 0.82; Recall = 0.79; AUC = 0.93; Accuracy = 88%
Kang [33]	Artificial Neural Network	CT/clinical/ laboratory	151	AUC = 0.95
Ho [34]	Convolutional Neural Network (ResNet50, Inception V3, DenseNet121	CT	297	Precision = 0.78; Recall = 0.80; AUC = 0.91; Accuracy = 93%
Weikert [45]	Convolutional Neural Network. Multiple CT metrics	CT/clinical/ laboratory	120	CT metrics alone, AUC = 0.88; Laboratory findings alone, AUC = 0.86; CT metrics and laboratory, AUC = 0.91
Fang [46]	Deep Learning, SVM, LR and RF	CT/clinical/ laboratory	193 (two data sets)	AUC = 0.813 (ICU)
Yan [47]	Deep Learning, (U-Net and 3D Convolution)	CT/Expert interpretation	221	AUC = 0.88 (ICU)
Proposed methodology	DL (Densenet-161), SVM, LDA	X-ray/CT and novel histogram design	596	Precision = 0.95; Recall = 0.95; Accuracy = 95%

## Data Availability

The data can be found using the information from references [36,37,38,39] in the following links: https://github.com/muhammedtalo/COVID-19; https://github.com/ieee8023/covid-chestxray-dataset; https://github.com/liuzhuang13/DenseNet.

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
