# Peer review of "Determination of the Severity and Percentage of COVID-19 Infection through a Hierarchical Deep Learning System"

_jpm, 2022, doi:10.3390/jpm12040535_

Round 1
Reviewer 1 Report
This article claimed a novel method based on hierarchical intelligent system, that analyze the application of deep learning models to detect and classify patients with COVID-19 using both X-ray and chest computed tomography (CT).
Unfortunately, we can hardly find any novelty in the said intelligent system. Please clearly explain the how the hierarchical intelligent system presented the novelty as compared to other existing system.
In addition, there is no comparison of result to other existing deep learning system, which is the main flaw of the paper.
Author Response
Reviewer #1
Comments and Suggestions for Authors
This article claimed a novel method based on hierarchical intelligent system, that analyze the application of deep learning models to detect and classify patients with COVID-19 using both X-ray and chest computed tomography (CT).
1.- Unfortunately, we can hardly find any novelty in the said intelligent system. Please clearly explain the how the hierarchical intelligent system presented the novelty as compared to other existing system.
2.- In addition, there is no comparison of result to other existing deep learning system, which is the main flaw of the paper.
Thank you so much for your kind review of our manuscript. Your comments were of great help and motivation to encourage our research.
Regarding the first question, the new version of the manuscript includes a new section (Section 3. Contribution of the proposed novel methodology) in which we have tried to highlight the novelty of this article by being able to automatically stratify the severity of the disease of a patient with COVID -19 by creating a histogram of the degree of infection in the different sections of the CT.
The new section has the following content:
“3. Contribution of the proposed novel methodology
In this paper, a novel method to detect and explore COVID -19 combining both X-ray and CT -scan images is presented. In addition to the detection of COVID by a CNN network using X-ray images, a CNN is used to investigate the degree of infection of each lung slice of the patients with a positive diagnosis using CT -scan images. The cohesiveness of the images allows us to examine the impact of the disease on the patient in a different way. Not only can it is possible to identify if a person is positive or not, but we can also see the condition of the lung sections. For these pre-trained CNNs in medical images, we are studying how different parameters (optimizers or the learning rate, according to additional work on other related studies) affect the effectiveness of the system.
But we do not stop at that point, because based on the predictions of the lung slices in the CT images, we use the SVM (supervised classification model) to classify the patient based on the severity of the infection. At this point, we make the greatest contribution by classifying based on the COVID -19 infection severity distribution of patients in a novel way for which there are few precedents in the literature. This is done by calculating the infection histograms of each patient based on the lung slices. To this end, we are creating a novel database of histograms focused on lung infections, a database that we have pre-labelled based on information about the infected slices. This can be used later and can serve as a starting point for other studies, in addition to classification tasks to locate the infection pattern that may be useful for studying the disease.
In this way and in a very visual way, the infection distribution can be visualized. The proposed novel methodology uses the information from all the CT slices in order to construct the histograms of COVID-19 infection. With these histograms, an SVM with LDA is trained to classify a patient in different states of severity.”
Regarding the second question, we included a new section with references to previously published contributions (Section 2). Papers have been presented that deals with the classification of patients with COVID -19, pneumonia and healthy patients, with intelligent systems that use medical images (X-Ray and CT). This subsection is called 2.1 Classification COVID -19 with X-ray and CT images. In addition, we have included another section with analyses recent contributions (although there are not too many) that use medical images to stratify the severity of a patient's disease (This section is called 2.2 Stratifying COVID -19 patients according to their severity).
We believe that the reviewer's comment is completely correct, calling and requesting a comparison between the methodology presented and other previous papers published in the bibliography. For this reason, a comparative table of the results obtained with the hierarchical system presented to determine the automatic and precise stratification of patients with COVID -19 into three categories: mild, moderate, and severe, has been presented in Table 6.
About the use of the English language, the whole paper has been comprehensively revised.
Reviewer 2 Report
- The research question(s) need to appear stronger and clearer.
- Identify the main findings and justify the novelty and contribution of the work. Please highlight the significance of your findings
- Please, clearly explain how your solution advances existing approaches,
- The author should introduce the proposed approach in more detail (in the abstract)
- The author should add a related works section to introduce some recent related works, discuss the motivation and limitations of previous works
- The author should compare the proposed algorithm with other recent works or provide a discussion. Otherwise, it's hard for the reader to identify the novelty and contribution of this work
- Results and parameters considered for evaluation have to be discussed in more details
- Does the proposed method have some shortcomings? In fact, shortcomings don’t reduce the availability of the proposed method. By contrast, it is a very suitable way to help readers to understand the proposed method comprehensively in my opinion.
- The language expression needs improvement. Please carefully check the manuscript and remove the typos.
Author Response
- The research question(s) need to appear stronger and clearer.
Thank you very much for this suggestion. An in-depth review of the article has been carried out, with the intention of improving its writing and the results obtained. The new ideas of this methodology are also highlighted, analyzing in detail the different phases of the proposed hierarchical structure. The novel system for generating histograms of CT images of patients with COVID -19 is analyzed in-depth to automatically stratify them into three levels. A major effort has also been made to present, compare, and analyze other methods presented in the bibliography. New sections and new tables have been included, and the entire text has been substantially improved.
- Identify the main findings and justify the novelty and contribution of the work. Please highlight the significance of your findings
Thank you very much for this suggestion. The paper includes a new section (Section 3. Contribution of the proposed novel methodology) in which we have tried to highlight the novelty of this article by being able to automatically stratify the severity of the disease of a patient with COVID -19 by creating a histogram of the degree of infection in the different sections of the CT.
The new section has the following content:
“3. Contribution of the proposed novel methodology.
In this paper, a novel method to detect and explore COVID -19 combining both X-ray and CT -scan images is presented. In addition to the detection of COVID by a CNN network using X-ray images, a CNN is used to investigate the degree of infection of each lung slice of the patients with a positive diagnosis using CT -scan images. The cohesiveness of the images allows us to examine the impact of the disease on the patient in a different way. Not only can it is possible to identify if a person is positive or not, but we can also see the condition of the lung sections. For these pre-trained CNNs in medical images, we are studying how different parameters (optimizers or the learning rate, according to additional work on other related studies) affect the effectiveness of the system.
But we do not stop at that point, because based on the predictions of the lung slices in the CT images, we use the SVM (supervised classification model) to classify the patient based on the severity of the infection. At this point, we make the greatest contribution by classifying based on the COVID -19 infection severity distribution of patients in a novel way for which there are few precedents in the literature. This is done by calculating the infection histograms of each patient based on the lung slices. To this end, we are creating a novel database of histograms focused on lung infections, a database that we have pre-labelled based on information about the infected slices. This can be used later and can serve as a starting point for other studies, in addition to classification tasks to locate the infection pattern that may be useful for studying the disease.
In this way and in a very visual way, the infection distribution can be visualized. The proposed novel methodology uses the information from all the CT slices in order to construct the histograms of COVID-19 infection. With these histograms, an SVM with LDA is trained to classify a patient in different states of severity.”
- Please, clearly explain how your solution advances existing approaches,
Thank you very much for this suggestion. A comparison between the methodology presented and other previous papers published in the bibliography is very relevant. For this reason, a comparative table of the results obtained with the hierarchical system presented to determine the automatic and precise stratification of patients with COVID -19 into three categories: mild, moderate, and severe, has been presented in Table 6.
- The author should introduce the proposed approach in more detail (in the abstract)
We totally agree with the reviewer. We have modified the abstract to indicate not only the different phases of the proposed novel hierarchical system, but also the relevance of stratifying patients with COVID -19 according to their severity (of great use for decision making by medical experts). The abstract is now:
“The coronavirus disease 2019 (COVID-19) has caused millions of deaths and one of the greatest health crises of all time. In this disease, one of the most important aspects is the early detection of the infection to avoid the spread. In addition to this, it is essential to know how the disease progresses in patients, to improve the patient care. This contribution presents a novel method based on hierarchical intelligent system, that analyze the application of deep learning models to detect and classify patients with COVID-19 using both X-ray and chest computed tomography (CT). The methodology was divided into three phases, the first being the detection of whether or not a patient suffers from COVID-19, the second step the evaluation of the percentage of infection of this disease and the final phase is to classify the patients according to their severity.
Stratification of patients suffering from COVID -19 according to their severity using automatic systems based on machine learning on medical images (especially X-Ray and CT of the lungs), provides a powerful tool to help medical experts in decision making. In this article, a new contribution is made to a stratification system in three severity levels (mild, moderate and severe) using a novel histogram database (which defines how the infection is in the different CT slices for a patient suffering from COVID -19).
The first two phases use CNN Densenet-161 pre-trained models, and the last uses SVM with LDA supervised learning algorithms as classification models.
The initial stage detects the presence of COVID-19 through X-Ray multi-class (COVID-19 vs. No-Findings vs. Pneumonia) and the results obtained for accuracy, precision, recall, and F1-score values are 88\%, 91\%, 87\%, and 89\%, respectively. The following stage manifested the percentage of infection COVID-19 in the slices CT-scans for a patient and the results in the metrics evaluation are 0.95 in Pearson Correlation coefficient, 5.14 in MAE and 8.47 in RMSE. The last stage finally classifies a patient in three degrees of severity in function of global infection at the lungs and the results achieved are 95\% in accuracy.”
- The author should add a related works section to introduce some recent related works, discuss the motivation and limitations of previous works
Thanks again for this comment, which undoubtedly improves the quality of the paper. For this reason, we have included a new section with references to previously published papers (Section 2), analysing the main advantages of these papers and their limitations (in terms of number of patients, information required, methods used, and results obtained). Papers have been presented dealing with the classification of patients with COVID -19, pneumonia and healthy patients, with intelligent systems using medical images ( X-rays and CT). This subsection is called 2.1 COVID -19 Classification with X-Ray and CT Images. In addition, we have included another section with analyses of recent contributions (although there are not too many) that use medical images to stratify the severity of a patient's disease (This section is called 2.2 Stratification of patients with COVID -19 according to their severity).
- The author should compare the proposed algorithm with other recent works or provide a discussion. Otherwise, it's hard for the reader to identify the novelty and contribution of this work
We believe that the reviewer's comment is completely correct, calling and requesting a comparison between the methodology presented and other previous papers published in the bibliography. Table 6 provides a comparative summary of various alternatives and methods presented in the literature for stratifying COVID -19 patients into different levels of disease severity. The results obtained with the methodology presented in this paper are included.
- Results and parameters considered for evaluation have to be discussed in more details
Thank you for that comment. We have tried to make it clear throughout the document describing the methodology and the parameters used (Sections 5 and 6), and it has been also discussed in detail.
- Does the proposed method have some shortcomings? In fact, shortcomings don’t reduce the availability of the proposed method. By contrast, it is a very suitable way to help readers to understand the proposed method comprehensively in my opinion.
We totally agree with this comment. It is indeed important to determine the limitations of a methodology. The main objection/limitation we can indicate is the need to use larger databases with a larger number of patients, where both the slices of CT and the patient itself are quantified/stratified. It is not easy to find such large databases in the literature, and this is of course a limitation.
- The language expression needs improvement. Please carefully check the manuscript and remove the typos.
The whole paper has been comprehensively revised.
Round 2
Reviewer 1 Report
Thanks for answering my previous doubt on the novelty. I have reviewed the paper, there exist some errors such as "toreduce". Besides, it is expected that authors can have some discussion on the limitations of the outcome as well as a detailed discussion on how this research outcome can be applied in practice.
Author Response
|
Journal |
JPM (ISSN 2075-4426) |
|
Manuscript ID |
jpm-1631419 |
|
Type |
Article |
|
Title |
Determination of the Severity and Percentage of COVID-19 Infection through a Hierarchical Deep Learning System |
|
Authors |
Sergio Ortiz * , Fernando Rojas , Olga Valenzuela , Luis Javier Herrera , Ignacio Rojas * |
Reviewer #1
Comments and Suggestions for Authors
Comments and Suggestions for Authors
Thanks for answering my previous doubt on the novelty. I have reviewed the paper, there exist some errors such as "toreduce". Besides, it is expected that authors can have some discussion on the limitations of the outcome as well as a detailed discussion on how this research outcome can be applied in practice.
Thank you so much for your kind review of our manuscript. Your comments were of great help and motivation to encourage our research, and as the review analyses, the discussion on the limitations of the methodology is very important.
We have modified some error (such as "toreduce").
In order to highlight the relevance of the limitations of the methodology, we have included a new Section. The Section is 7. Limitations and practical applications, with the following information:
“7. Limitations and practical applications
One of the most important shortcomings in the application of deep learning in medicine, and focusing on the new disease of COVID-19, is the size of the data set and the decentralization of this information. When the data set is too small, the images may have natural features too fixed, which could interfere with the model. This effect is responsible for the appearance of a bottleneck and therefore it is recommended that the data be in large amounts (Fig. 14). For this reasons and due to the limited number of COVID-19 images, more experiments are needed on a larger set of clearly labeled COVID-19 images for a more detailed estimation of the accuracy of the studies done in this paper. It is also important that the data distributions have a similar density for each class, although this can be mitigated by studying not only aspects such as accuracy but also markers such as F1-scored of the isolated classes or by applying up-sampling mechanisms through partial image alteration techniques.\\
Figure. 14
Another of the limitations that are being considered in the implementations is related to the scarcity of additional patient information. The image sets used have the absence of metadata on the pathologies present in the subjects, age, sex or other information that might be necessary to detect possible bias. We consider that not having information about factors such as the pulmonary history of the patients is a great impediment. Some of the patients may have a previous lung problem in this case of COVID-19 and do not consider this task due to this lack of information. In addition to this, it is also very important to know information that can emphasize the seriousness of COVID-19. Having certain information about the temporal evolution of the disease can also be of great importance to study how the stage of the disease influences the results of the different models.
In the applications of deep learning models, it is possible to exceed the human precision of many specific tasks within the medical field. However, small variations in the form data acquisition for these models can lead to wrong decisions. This calls into question both the models and their reliability, without being tested for large amounts of data and for highly variable data. Moral and critical questions are also raised that can be questioned by people about the creation of medical prediction models by people not specialized in the medical subject in question.
For the practical implementation of this research, it is necessary to address and study all the possible external influences that can cause fluctuations in model prediction and explainability, to understand and trust the results and output generated by algorithms. In addition to this and as the main limitation already mentioned, it would be necessary to collect a global database of COVID-19 from different sources and analyze the model and the results in detail. In short, through the work carried out, an attempt is made to provide the community with a new approach from which its study can be continued so that the current COVID detection system can be improved.”